# Prevalence of Anemia and Iron Deficiency in Women of Reproductive Age in Cuba and Associated Factors

**DOI:** 10.3390/ijerph20065110

**Published:** 2023-03-14

**Authors:** Gisela María Pita-Rodríguez, Beatriz Basabe-Tuero, María Elena Díaz-Sánchez, Karen Alfonso-Sagué, Ana María Gómez Álvarez, Minerva Montero-Díaz, Sonia Valdés-Perdomo, Cristina Chávez-Chong, Ernesto Rodríguez-Martinez, Yoandry Díaz-Fuentes, Elisa Llera-Abreu, Ahindris Calzadilla-Cámbara, Israel Ríos-Castillo

**Affiliations:** 1National Institute of Hygiene, Epidemiology and Microbiology (INHEM), Havana 10300, Cuba; 2Department of Physiology Science, Latin American School of Medicine, Havana 11600, Cuba; 3Institute of Cybernetics, Mathematics and Physics (ICIMAF), Havana 10400, Cuba; 4Food and Agricultural Organization (FAO), Sub Regional Office in Mesoamerica, Panama City 0843-00006, Panama; 5Nutrition and Dietetic School, University of Panama, Panama City 3366, Panama

**Keywords:** anemia, iron deficiency, inflammation, women of reproductive age, overweight and obesity

## Abstract

This study aims to evaluate the prevalence of anemia and iron deficiency in women of reproductive age and the association with inflammation, global overweight, adiposity, and menorrhagia. A sample design of women of reproductive age from the Eastern, Central, and Havana Regions was carried out. Biochemical determinations of hemoglobin, serum ferritin, soluble transferrin receptors, leukocytes, C-reactive protein, alpha-1 acid glycoprotein, and homocysteine were performed. Serum ferritin was also adjusted by inflammation. Nutritional status was assessed, and menstrual characteristics were collected by survey. A total of 742 women were studied. The prevalence of anemia was 21.4%, iron storage deficiency at 16.0%, and erythropoietic dysfunction at 5.4%, with inflammation at 47.0% and elevated homocysteine at 18.6%. Global overweight was 46.2% and increased adiposity at 58.4%. Anemia is associated with iron deposition deficiency (OR = 3.023 (1.816–5.033)) and with erythropoietic deficiency (OR = 5.62 (3.03–10.39)), but not with inflammation, global overweight, and adiposity. Global overweight was found to be associated with inflammation (OR = 2.23 (1.41–3.53)). Anemia was associated with heavy menstrual bleeding (OR = 1.92 (1.34–2.76)). Homocysteine was associated with inflammation (OR = 2.05 (1.08–3.90)), but not with anemia. In conclusion, anemia in Cuba is classified as a moderate public health problem, but not iron deficiency. A high prevalence of overweight and obesity was found, associated with inflammation, but not with anemia or iron deficiency. Heavy menstrual bleeding is a factor associated with anemia.

## 1. Introduction

Anemia in women of reproductive age (WRA) is a worldwide public health problem that remains a nutritional challenge [1], and there is a decrease of only four percentage points from 1995 (33%) to 2011 (29%) [2]. Factors associated with anemia have been reported as regular blood loss (due to menstrual bleeding), pregnancy-related complications, depletion of iron stores, and therefore, increased requirements [3]. The double burden of malnutrition consists of malnutrition due to micronutrient deficiency and obesity, which can be found in the individual as well as in the family and in populations [4]. Among the nutrient deficiencies, a relationship between iron deficiency and obesity has been observed [5].

In medical practice, inflammation is a common cause of anemia, but the magnitude of its attributable causation in developing countries is unknown [6]. Inflammation is a sign of the organism responding to infections with seasonal bacteria or viruses [7]. Thus, inflammation is an important trigger for chronic diseases and also for obesity [8]. During the inflammatory process, interleukins are secreted that stimulate the secretion of hepcidin, a known hormonal peptide released from the liver. Hepcidin acts on intestinal cells, preventing the absorption of iron from food, and on the phagocytic mononuclear system, thus sequestering circulating iron in the blood [9,10]. In these cases, iron deficiency is due to both low absorption caused by poor diet and to low use by the body.

On the other hand, obesity is characterized by systemic, chronic, and low-grade inflammation, with an increase in C-reactive protein (CRP), Alpha 1-acid glycoprotein (AGP), and interleukin 6 (IL-6) [9,10,11]. Thus, malnutrition is related to iron deficiency and anemia through hepcidin secretion stimulated by high levels of IL-6 during inflammation [10]. In developing countries, the double burden of malnutrition and infections are increasing [4,12]. Therefore, the evaluation of the potential relationship of these factors with serum ferritin, as a protein for the evaluation of iron nutritional status, is important to explain the high prevalence of iron deficiency and its relationship with anemia [8].

Based on the knowledge of the metabolic pathway of homocysteine (Hcy) [13] and because of the difficulties in assessing serum or erythrocyte folate status, the use of Hcy metabolite as an indirect indicator of the nutritional status of these vitamins in the individual has been proposed [14]. Menstruation is a physiological process that can produce heavy menstrual bleeding (HMS), which interferes with a woman’s physical, social, emotional, and/or material quality of life, and which may or may not be accompanied by other symptoms [15]. Heavy menses is a common condition, affecting one in four WRA [16]. The group of WRA represents a very important link in the life cycle, but it has not been a population deeply studied in Cuba; their nutritional status is of great importance for a healthy pregnancy, a child with adequate birth weight, and adequate iron status. Therefore, it is necessary to know the factors that may be influencing anemia and iron deficiency in this group to propose actions. This study aimed to evaluate the prevalence of anemia and iron deficiency in women of reproductive age and its association with inflammation, global overweight, and volume of menses.

## 2. Materials and Methods

The subject universe consisted of mothers of children aged 6–59 months studied by the national survey of anemia and iron deficiency in Cuban preschool children [17], conducted by the National Institute of Hygiene, Epidemiology, and Microbiology (INHEM acronym in Spanish) between February and April of each year during 2016–2018.

Inclusion criteria were healthy WRA, aged 18–40 years, mothers of children from the Eastern Region (Santiago de Cuba and Holguin), Central Region (Sancti Spiritus and Cienfuegos), and Havana. Exclusion criteria were pregnant women, postpartum or with a delivery time of fewer than six months, and women with sickle cell disease or seen in consultations for hematological disorders. In addition, we excluded those women with acute illnesses, or those suffering from chronic diseases such as cancer, diabetes mellitus, arterial hypertension, hypothyroidism, hyperthyroidism, severe asthma, chronic obstructive pulmonary disease, or renal insufficiency.

### 2.1. Study Variables

Table 1 shows the biochemical and anthropometric indicators with the cut-off points used. Anemia was defined as hemoglobin < 120 g/L, and iron deficiency was defined as ferritin concentration < 15 μg/L, the cut-off points recommended by WHO [9,10]. The nutritional status was assessed to evaluate overweight and obesity. Weight, height, and waist circumference were measured. In the analysis of the anthropometric results, the cases belonging to the “overweight” and “obese” groups were classified as “global overweight”.

As risk factors for maternal anemia, the quality and quantity of menstruation were recorded with a questionnaire. In the menstrual survey, questions 5, 6, 8 and 9 of the “Survey of Symptoms during Your Monthly Period” [24] were adapted, and the practical clinical guidelines of the Society of Obstetricians and Gynecologists of Canada [23] were also used. The questions include description of the menstrual period (question 5); duration of the menstrual period (question 6); if the participant has seen a doctor because of vaginal bleeding (question 8); and if the participant has had to attend the emergency room for vaginal bleeding (question 9). The survey on menstrual bleeding or menorrhagia in women of childbearing age has been applied by other authors [25]. The survey includes questions about the amount of bleeding; bleeding frequency; duration of bleeding; regularity of the menstrual cycle; non-menstrual bleeding; and need for medical attention for bleeding.

### 2.2. Biochemical Data

Blood was extracted by antecubital puncture after fasting for at least eight hours, before the questionnaire was administered, and after signing the consent form. Five mL of whole blood, 1 mL with 10% EDTA for Hemoglobin (Hb) and leukocytes determination, and 4 mL for serum collection were extracted. On the day of the extraction, the samples were centrifuged, and the serum was stored at −40 °C until the time of analysis for ferritin, sTfR, Hcy, and inflammation indicators

Hb and leukocytes were determined with an ABX Micros 60 Hematology Counter (Horiba, France). Iron deficiency was measured by ferritin and soluble transferrin receptor (sTfR) concentration, and inflammation by serum high-sensitivity C-reactive protein (CRP-hs) and α-1 acid glycoprotein (AGP). Ferritin and inflammation indicators were determined by the immunoturbidimetric method (CPM Scientifica Tecnologie Biomediche, Rome, Italy) using INLAB 240. Ferritin concentrations were adjusted using the quantile regression method [17,26,27].

The sTfR was performed by the ELISA method, Ramco Laboratories INC [28]. The sTfR was estimated in a subsample of Cienfuegos and Havana, and Hcy in a subsample of the eastern provinces and Havana. Hcy was determined by the enzymatic method (CPM Scientifica Tecnologie Biomediche, Rome, Italy) with INLAB 240. Homocysteine was used as an indirect indicator of folate deficiency. All cutoff points used are found in Table 1. The determinations were carried out at the INHEM’s Nutritional Anemia Laboratory.

### 2.3. Statistical Analysis

Biochemical variables were described according to their distribution by percentiles (median, 25th percentile, and 75th percentile). For categorical variables, prevalence and 95% confidence intervals (95% CI) were calculated. Prevalence calculations were performed taking into account the sample design. Serum ferritin was adjusted by inflammation using quantile regression, which is a natural extension of the standard regression model and allows separate regression models to be used for different parts of the dependent variable’s distribution. Quantile regression’s additional flexibility may broaden the description of inflammation’s effect on ferritin’s conditional distribution. An additional advantage to quantile regression is that it does not depend on normality assumptions or transformations. The quantile regression methodology has already been used in the correction of ferritin levels in preschool children in Cuba [17,26]. In the association analysis, the variables were categorized into two levels, and the odds ratio (OR) and 95% CI were used. In this case, for the estimations, the sample design was also taken into account. For the analysis of the menstrual volumes of WRA, they were grouped into two groups, according to pad condition: Soaking through + Strongly wet/Moderate + Slightly wet. SPSS v20.0 statistical software was used for database preparation. The SAS 9.1 statistical package was used to take into account the complex sampling design of the sample for the estimation of the statistics and their standard errors as well as the statistical tests used. The SPSS program was used to evaluate the associations. A significance level of *p* < 0.05 was considered in all cases.

### 2.4. Ethical Aspects

The project was approved by the ethics committee of the INHEM and authorized by the Maternal and Child Division of the Ministry of Public Health of Cuba. Written informed consent was obtained from the mothers of the children participating in the study. At each stage, the agreements of the World Medical Association in the Declaration of Helsinki on ethical principles for medical research in humans were taken into account, and compliance with the basic principles of all research with human beings was monitored [29].

## 3. Results

The final sample of WRA was 742 women, of which 729 were samples with usable Hb and leukocytes, and 711 with ferritin and inflammation. The cause of the missing data does not represent a risk of bias, since it responds to random causes such as coagulation of the whole blood sample or not enough serum for all determinations. A subsample of 249 was determined for sTfR and 381 for Hcy.

### 3.1. Anemia and Iron Nutritional Status

Anemia was found in 21.4% of the women studied (Table 2); the majority (81.1%) was classified as mild, with 18.1% moderate, and 0.8% severe anemia. Iron storage deficiency was found in less than one-third of the cases and classified as a mild public health problem. The prevalence of erythropoietic dysfunction was low (5.4%). Most ferritin values were in the reference range of 15–150 µmol/L 82.7% (74.2–88.8), but seven cases were found with values above this range, indicative of risk of iron overload.

Inflammation, both acute and chronic, was found in about one-third of all women and general inflammation in about half of the sample, but not leukocytosis, which was found in less than one-tenth. The prevalence of elevated Hcy was found in about one-fifth of the sample, resulting in a red flag in the study for this indicator (Table 2).

### 3.2. Nutritional Status

In the nutritional evaluation of women, 42.8% (39.5–46.1) had adequate nutritional status, 11% (7.8–14.2) were undernourished, and 46.2% were with global overweight (overweight 30.4% (25.3–35.5), and obese 15.8% (13.0–18.7). The degree of central adiposity was adequate in 58.4% (52.9–63.9), with increased risk in 22.8% (15.4–30.2), and very increased risk in 18.7% (14.5–23.0).

### 3.3. Menstrual Analysis

Of the WRA, 19.8% (14.4–25.3) reported soaking through the sanitary pad, and 22.7% (18.4–26.9) had very wet pads. The other group reported moderately wet, 40.2% (31.2–49.2), and slightly wet, 17.2% (9.3–25.1). Only one case reported that menstruation was variable in quantity. Most of the women—87.7% (85.5–89.9)—reported that the frequency of menses was normal (24–38 days), as was the duration (3–8 days) for 93.6% (91.7–95.5). Acute bleeding was reported by 7.4% (4.6–10.2), and 13.2% (9.5–16.9) went to the doctor for this cause.

### 3.4. Association Analysis

Anemia in women of reproductive age was associated with iron storage deficiency (OR = 3.02 (1.82–5.03)) and with erythropoietic deficiency (OR = 5.62 (3.03–10.39)), but not with inflammation (OR = 1.00 (0.65–1.54)), global overweight (OR = 0.80(0.57–1.12)), and central adiposity (OR = 0.80 (0.57–1.12)). Iron storage deficiency explained 29.8% of the anemia found. Iron storage deficiency was not associated with global overweight (OR = 0.71 (0.48–1.06)), but it was associated with central adiposity (OR = 0.59 (0.38–0.91)) as a protection factor. Global overweight in women was found to be associated with inflammation (OR = 2.23 (1.41–3.53)), mainly with elevated CRP (OR = 3.06 (1.89–4.94)) rather than with AGP (OR = 1.80 (1.05–3.08)). Adiposity and inflammation behaved in a similar way, with (OR = 3.23 (2.32–4.51)), with higher values for CRP (OR = 3.76 (2.29–6.16)) than those for AGP (OR = 1.94 (1.19–3.15)). Anemia in women was explained by the higher volume of menstruation (Soaking through + Strongly wet/Moderate and Slightly) (OR = 1.92 (1.34–2.76)) and heavy menstrual bleeding, which explained 55.4% of anemia in women, but did not explain iron storage deficiency as well (OR = 1.35 (0.80–2.27)). However, the volume of menstruation is not accurate, and it is subject to measurement failures. The analysis showed that Hcy was also not associated with anemia (OR = 0.94 (0.44–2.00)), but it was associated with inflammation (OR = 2.05 (1.08–3.90)), by elevated CRP (OR = 1.17 (0.85–1.60)) and mostly by elevated AGP (OR = 1.65 (1.07–2.55)). However, the association with global overweight and adiposity was not relevant (OR = 1.02 (0.73–1.42)) and (OR = 1.32 (0.74–2.33)), respectively.

## 4. Discussion

Anemia in the studied WRA had a similar prevalence to that found in 2008 (n = 1802) in women from the eastern provinces (19.9%) [30]. In our study, WRA had adequate ferritin values, and less than 20% showed iron storage deficiencies that did not affect erythropoiesis. These data are striking because in the study performed on 391 WRA in Havana in 2017, the iron deficiency, adjusted for inflammation, was found to be 68%. Anemia was associated with iron deficiency and erythropoietic dysfunction with greater strength than that obtained in the Havana study [31], and no positive association was found with inflammation. Leukocytosis was low, unlike the results of the WRA group in Havana; however, indicators of subclinical inflammation in this study are higher than those found in 2014 (CRP 8.4% and AGP 19.9%) [31].

Stevens et al. [32] performed a national, regional, and global estimation of the severity of anemia in 133 countries, in WRA aged 15–49 years. They reported an anemia prevalence of 30% (27–33%), with a decrease in severe and moderate anemia, but considered that this progress is insufficient to reach the proposed global goals for 2025 (to reduce anemia by 50% according to each country’s baseline) [33]. The decrease in anemia by decade was mostly in the Latin American and Caribbean region. The reported prevalence was higher than those found in our study.

Kinyoki et al. [34] conducted a geospatial study of anemia prevalence estimates in women of reproductive age (15–49 years) from 2000 to 2018 in 82 low- and middle-income countries. The results showed moderate improvements in the overall prevalence of anemia in the various countries, but only three countries (China, Iran and Thailand) could be identified as having a high likelihood of achieving the proposed targets at the national level. Wirth et al. [35] conducted a study in Somalia between 2018 and 2019 in 583 women of reproductive age, where they explored, in addition to anemia and iron deficiency, vitamin A deficiency, finding that the main risk factor for anemia was iron deficiency.

No studies have been found for Cuba where folate concentrations in women of reproductive age have been estimated. There are only dietary studies indicating folate intake deficiency in preschool children and pregnant women, which were obtained by the nutritional surveillance system established in the country [36]. Rogers et al. [14] conducted an analysis of 45 relevant studies in 39 countries to estimate folate deficiencies in WRA between 2000 and 2014. The deficiency found was greater than 20% in most low-income countries, in contrast to high-income countries, where it was less than 5%. In early stages of folate deficiency, homocysteine concentrations are elevated and appear even at elevated serum or plasma folate levels (10 or 14 nmol/L) [37]. Evaluation of homocysteine in WRA in this study was not useful for estimating folic acid deficiency [13]. However, it evidenced its association with inflammation; therefore, it is a factor to keep in mind in this population, and the elevated values in the subsample analyzed are not negligible. Rosabal Nieves [13] recognized several hereditary, pathological, nutritional, and pharmacological events capable of inducing hyper-homocysteinemia. Sex and age are the most important physiological causes of elevated plasma homocysteine. Serum tHcy levels increase with age, due in part to physiological decline in renal function. However, we did not evaluate the relationship between age and homocysteine. Therefore, it is necessary to continue investigating the relationship of anemia and iron deficiency with other biochemical markers, such as folate.

Although the level of zinc was not evaluated in this study, the literature suggests that zinc can be related with anemia and iron deficiency. The recent study of Zn status conducted in a sample of 654 women of reproductive age evidenced that Zn deficiency was high, at 35.7%, and 25.0% of Zn-deficient women had anemia. These results suggest an association between anemia and Zn deficiency not previously studied in Cuba [38]. Greffeuille et al. [39] evaluated Zn and Hb concentrations in 22,633 WRA from nationally representative studies from different countries and found a prevalence of Zn deficiency between 9.8 and 84.7%. The prevalence of Zn deficiency was greater than 20% in most countries. Zn concentration had a positive association with Hb in about half of the countries, independent of iron status, and was significantly related to anemia in most countries, thus concluding that strategies to combat Zn deficiency may help reduce the prevalence of anemia. In humans, no direct relationship between Zn concentration and red cell production has been evidenced, but Zn is directly involved in erythropoietic differentiation and development [40]. Therefore, more studies are required to evaluate the relationship of anemia and iron deficiency with micronutrient deficiency, obesity, and NCDs.

In the WRA of the present study, malnutrition had high figures, with a predominance of global overweight and a high percentage of adiposity, but neither of these was associated with anemia or iron deficiency. This result coincides with that of Havana, where excess body weight was not positively associated with anemia or iron deficiency, but central adiposity was a protective factor for both cases [31]. In a regional study in Latin America in 3254 women aged 15 to <45 years, Herrera-Cuenca et al. [41] found that 58.7% were overweight, associated with adequate iron intake. Christian et al. [42] conducted a national study in Ghana in 1063 women aged 15–49 years and found that the prevalence of overall overweight was 39%, anemia 22%, and deficiency of a micronutrient 62%.

The joint occurrence of overall overweight and anemia was 6.7%, and with deficiency of at least one micronutrient 23.6%. Kamruzzaman [43] conducted a study in Bangladesh, using nationally representative data on 5680 women aged 15–49 years, where he found that the probability of anemia was higher in the undernourished than in the overweight and obese. Although it is known that nutritional deficiencies are associated with anemia, the relationship between obesity, inflammation, and anemia has also been described in the literature [6,7]. Therefore, its study is important. Despite this, in this study, we did not observe a relationship between obesity and anemia or iron deficiency.

These results are in agreement with those found in the current study, where anemia and iron deficiency were not associated with global overweight or adiposity, which, on the contrary, proved to be protective factors. The subjective assessment of women’s menstrual volume proved to be a rough indicator of how much anemia may be caused by menstrual bleeding, as more than half of the anemia can be explained by this cause.

In a pilot study conducted in 44 adult African-American women by Bernardi et al. [44] in 2016, almost half reported heavy and very heavy menses; they found anemia in 18.2% and iron deficiency in 69.2%, with significant association between anemia and perceived menstrual volume. Among those who reported high menstrual volume, 35% had anemia. These figures are lower than those found in this work. Kiran et al. in 2018 [16] report that in England and Wales, an estimated 50,000 women with very heavy menses are transferred annually from primary care to secondary care services of gynecology in the National Health Service, and menstrual disorders occur in 20% of women, affecting their quality of life.

Ding et al. [45] in 2019 studied the prevalence and risk factors of 2356 WRA (18–50 years) in a Chinese community who experienced heavy menses and assessed its effect on daily life. They observed that 18.2% of them reported heavy menses, and this was significantly associated with iron deficiency anemia (OR = 1.56 (1.06–2.29)). In the current study, heavy menstruation was associated with anemia, but not with iron deficiency. In contrast to our findings, using SF values adjusted for inflammation, Zazo et al. [46] studied the causes of iron deficiency anemia prevalence in 161,315 women of reproductive age 18–35 years, seen in Madrid from 2010–2015. They highlighted that 61.8% of iron deficiency anemia, which had no known associated cause, was due solely to menstrual losses [46]. Anemia is a multifactorial disease, and there may be other associated factors that were not evaluated in this study.

The limitations of the availability of iron for the erythroid progenitors during an infectious process are one of the central causes of anemia due to inflammation [7]. Mild iron deficiency is associated with protection from certain infections, thus preventing morbidity and mortality from infectious diseases, particularly in children. On the other hand, severe iron deficiency has a negative impact on the proliferation of immune cells and therefore weakens the immune response [47,48]. In places with high rates of infections, other factors for anemia are bleeding episodes of gastrointestinal origin, parasitism or urogenital losses, and menstruation disorder. These subjects develop absolute or relative iron deficiency. As a consequence, iron is not available for essential metabolic processes, including cellular respiration.

Anemia in WRA is an important public health problem worldwide and in Cuba, mainly due to the increased risk of anemia during pregnancy and the negative effects on infants [33,36,38]. However, iron deficiency anemia is the last stage of nutritional iron deficiency [49]. Therefore, evaluating iron deficiency is crucial to avoid the consequences of anemia early on. Anemia prevention policies and programs require the generation of timely scientific evidence on the iron nutrition situation in the population. The present study on anemia and iron deficiency in women of childbearing age also allows us to identify the potential relationships that occur with malnutrition due to excess weight and menstrual disorders.

Some limitations in our study are the design (cross-sectional), so that it was impossible to examine the temporality of the indicators with other factors; second, that it was not possible to study the WRA of the western provinces, which implies that the results may be influenced by the lack of these data. The third limitation is that it was not possible to study other important nutritional causes of anemia in the vulnerable population, such as folic acid deficiency.

## 5. Conclusions

Anemia in WRA in Cuba is classified as a moderate public health problem, but iron deficiency is a mild public health problem. Women of reproductive age have a high prevalence of overweight and obesity that is not associated with anemia or iron deficiency but is associated with inflammation. Heavy menstrual bleeding is a factor associated with anemia.

## Figures and Tables

**Table 1 ijerph-20-05110-t001:** Variables used for the evaluation of women of reproductive age. Cuba 2016–2018.

Variables	Biomarker and Cut-Off Point
Anemia [18]	Hb < 120 g/L
Severity of anemia at individual level [18]	Severe: Hb < 70 g/L
	Moderate: 70–99 g/L
	Low: 100–119 g/L
Anemia as public health problem classification at population level [19]	Severe: anemia prevalence ≥ 40%
	Moderate: anemia prevalence 20–39.9%
	Low: anemia prevalence 5–19.9%
	No public health problem: anemia prevalence < 5%
**Fe deficiency [19]**	
Depleted Fe stores	Serum ferritin (SF) < 15 µg/L ^2^
Normal Fe stores	Serum ferritin ≥ 15–150 µg/L
Risk of Fe overload	Serum ferritin ˃ 150 µg/L
Fe deficiency as public health problem using SF levels [19]	Severe: prevalence of depleted Fe stores ≥ 40%
	Moderate: prevalence of depleted Fe stores 20–39.9%
	Low: prevalence of depleted Fe stores 5–19.9%
	No public health problem: prevalence of depleted Fe stores < 5%
**Transferrin receptor (sTfR) ^1^**	
Erythropoietic dysfunction due to iron deficiency	>8.3 μg/mL
Without erythropoietic dysfunction due to iron deficiency	≤8.3 μg/mL
**Leukocytes** (ABX Hematologie Micro 60-OT, ABX Diagnostics, France)	
Leukocytosis	>10,000/mm ^3^
Normal values for leukocytes	≤10,000/mm ^3^
Inflammation [19]	C-reactive protein (hs-CRP): >5 mg/L
	alpha-1-acid glycoprotein (AGP): ˃1 g/L
Elevated plasma homocysteine in women ^3^ [13]	˃15.0 µmol/L
**Anthropometric**	
Nutritional status	
BMI Kg/m^2^ [20,21]	Low weight: <18.5
	Normal weight: 18.5–24.9
	Overweight: 25–29.9
	Obesity: ≥30.0
	Excess weight: ≥25
**Adiposity** [22]	
Waist circumference (cm)	No risk: ≤80 cm
	Increased risk: >80 cm
	Very increased risk: >88 cm
**Epidemiological variables**	
Types of menstruation [23]	
- Menstrual volume	Soaks through the sanitary napkin
	Strong soaked sanitary napkin
	Moderate soaked sanitary napkin
	Low soaked sanitary napkin
	Normal (24–38 days)
- Frequency	>38 days
	<24 days
	NA
- Duration	Normal (3–8 days)
	>8 days
	<3 days
	variable
- Acute bleeding	Yes
	No
- Do you go to the doctor for acute bleeding?	Yes
	No

^1^ The reference values are those referred to in the diagnostic kit. ^2^ Ferritin values were adjusted for inflammation, as described elsewhere, using quantile regression [18,19]. ^3^ Hcy was used as an indirect biomarker of folate deficiency [13].

**Table 2 ijerph-20-05110-t002:** Biomarkers and prevalence of anemia, iron deficiency, and inflammation in women of childbearing age, Cuba 2016–2018.

Variables	n	Unit	P25 ^1^	P50 ^1^	P75 ^1^	Min ^1^	Max ^1^	Prevalence (CI95%) ^1^
Hemoglobin	729	g/L	123	130	136	64	154	
Anemia (Hb < 120 g/L)	729	Yes						21.4% (15.6–27.2%)
Serum Ferritin (SF)	711	µg/L	19.6	33.7	62.6	3.63	1001	
Depleted Fe stores (SF < 15 µg/L) ^2^	711	Yes						16.0% (9.9–24.8%)
Transferrin receptor (sTfR)	249	μg/mL	3.52	4.18	5.25	2.21	31.2	
Erythropoietic dysfunction due to iron deficiency (sTfR ˃ 8.3 μg/mL)	249	Yes						5.4% (0.0–11.6%)
Leukocytes	729	mm^3^	5.9	7.1	8.5	2.6	17.7	
Leukocytosis (>10,000/mm^3^)	729	Yes						8.3% (4.4–12.2%)
C-reactive protein (hs-CRP)	711	mg/L	1.14	2.42	6.43	0.1	86.8	
Inflammation by (hs-CRP > 5 mg/L)	711	Yes						31.0% (27.0–34.8%)
alpha-1-acid glycoprotein (AGP)	711	g/L	0.64	0.82	1.0	0.18	2.88	
Inflammation by (AGP ˃ 1 g/L)	711	Yes						31.9% (22.0–41.7%)
Adjusted inflammation								47.0% (38.8–55.0%)
plasma homocysteine (Hcy)	381	µmol/L	9.7	12.0	13.8	1.0	57.0	
Hcy increased (˃15 µmol/L)	381	Yes						18.6% (8.7–35.4%)

^1^ P25 = percentile 25; P50 = median; P75 = percentile 75; Min = minimum value; Max = maximum value; CI 95% = confidence interval 95%. ^2^ Serum ferritin was adjusted by inflammation, using quantile regression [17,18,19,26,29].

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
