# Peer review of "Prevalence of Anemia and Iron Deficiency in Women of Reproductive Age in Cuba and Associated Factors"

_ijerph, 2023, doi:10.3390/ijerph20065110_

Round 1

Reviewer 1 Report

This study aims to evaluate the prevalence of anemia and iron deficiency in women of reproductive age and the association with inflammation, global overweight, and menorrhagia

Here are suggestions for fixes.

Suggest of title: Prevalence of anemia and iron deficiency in women of reproductive age in Cuba and associated factors

1.Introduction

Between lines 43 and 45 is mentioned: "In the world, iron deficiency and obesity
 coexist, known as the double burden of malnutrition, which can be found in the individual as well as in the family and populations". The double burden of malnutrition consists of malnutrition due to micronutrient deficiency and obesity. It doesn't just imply iron deficiency. Therefore, the sentence should be revised. Among the nutrient deficiencies, the relationship between iron deficiency and obesity has been observed. 

Line 43: In the world, nutritional deficiency and obesity(...)

Justify in the introduction why researching inflammation, overweight and menstrual health in women of reproductive age is important.

2nd Materials and Methods

Line 69: delete under 18 and over 40 years old because you don't include this.

Line 76: Specify at which time blood collection occurred.  Was it a fasting  blood collection?  It was before or after the questionnaire was applied? Before or after signing the consent form?

Line 79: mention that the cutoff points are found in table 1

Line 93: suggestion: the topic of study variables could come before Biochemical data

Line 102: describe questions 5,6,8 and 9 of the "Survey 102 of Symptoms during Your Monthly Period

Line 113: this methodology has already been used by other authors or is it standardized?

I suggest better describing the instrument used  in data collection.

3. Results

Line 134: You have placed anemia within the topic iron nutritional status. What mechanism did you use to say that this anemia is due to iron deficiency? I suggest renaming to Anemia and iron nutritional status.

Line 144 and 147: align the title and legend in the table

Line 161. topic 3.3 is repeated

Line 167 – mentions that the deficiency of iron reserves was not associated with central adiposity, but in line 167 it is mentioned that the OR=0.59 (0.38-0.91). That is, there was an association.

Line 165 comments that iron deficiency explained only 29.8% of the anemia found. Removing the word only, because 29.8% is a very important and considerable percentage.

Line 171 – addresses   that the issue of the highest volume of menstruation and heavy menstrual bleeding explained 55.4% of anemia in women, but had no association with iron reserves.  To consider that this analysis of the volume of menstruation is not accurate and is subject to measurement failures.  Or mention references validating this approach.

According to WHO (2020), excerpt that I reproduce in full below:

Determination of both CRP and AGP concentrations may be important because they reflect different phases of the acute phase response that range from acute infection to chronic inflammation.
• Possible adjustments include the following:
o the higher ferritin cut-off adjustment approach uses a higher ferritin-concentration cut-off value for individuals with infection/inflammation, e.g. <30 μg/L;
o the exclusion approach uses the inflammation, malaria-biomarker information, or both, to exclude individuals with elevated inflammation (as defined by a CRP concentration >5 mg/L, AGP concentration >1 g/L, or both) or individuals with malaria infection;

o the arithmetic correction factor approach applies an arithmetic correction factor by grouping inflammation into groups, e.g. (i) reference (both CRP concentration <5 mg/L and AGP concentration<1 g/L); (ii) incubation (CRP concentration >5 mg/L and AGP concentration <1 g/L); (iii) early convalescence (both CRP concentration >5 mg/L and AGP concentration >1 g/L); and (iv) late convalescence (CRP concentration <5 mg/L and AGP concentration >1 g/L)”

The authors performed c-reactive protein and acid alpha-glycoprotein analyses. Therefore, this adjustment should also be presented.

The reference of this publication is mentioned in the article submitted in reference 10, but this analysis was not applied.

Table 2 mentions that 16% (9.9% -24.8%) presented iron reserve depletion using ferritin (SF< 15 μg/L). However, in this analysis, women with inflammation evaluated by C-reactive protein (hs-CRP > 5mg/L) were not excluded, while the presence of inflammation was 31% (27-34.8%). Ferritin is known to be a reactant to the acute phase and increases in the presence of infection/inflammation.

  According to WHO, it is recommended to adjust ferritin to SF< 30 μg/L for individuals with infection/inflammation.

4. Discussion

  Line 181 – mentions that WRA had adequate ferritin values and less than 20% showed iron storage deficiencies that this does not affect erythropoieses.

Line 183:  mentions "This data is striking because in the study performed on 391 WRA in Havana, the iron deficiency adjusted for inflammation was found to be 68%."

  The data presented in Table 2 of 16% (9.9% -24.8%) were not adjusted for inflammation.  Therefore, it should also present the data using the cutoff point of SF< 30 μg/L for women with C-reactive protein or altered acid alpha-glycoprotein. In this same table, data are presented that  31%  presented inflammation, because they  had  altered C-reactive protein and 31.9% with AGP > 1g/L.

The fact that it did not identify an association between anemia and inflammation does   not justify not performing ferritin adjustment by inflammatory indicators.

Line 212 must have occurred typo. Exchange homocystine for homocysteine.

Line 218 – mentions that sex and age are the most important physiological causes of elevated plasma homocysteine".  However, it would be interesting to mention whether or not this research was associated with these variables.

Lines 233 to 235 comment that there was no association of the global overweight and that there was no association with iron deficiency.   As discussed above, the values for iron deficiency assessment should be reviewed.  Therefore, further analysis should be performed to assess whether there is an association between global overweight and iron deficiency.

Lines 263-264 – Mentions that heavy menstruation was not associated with iron deficiency.  Review classification with iron deficiency and do new analysis with heavy menstruation. The study by Zazo et al (reference 35) lines 265 to 267 showed that 61.1% had iron deficiency anemia.

Lina 268 - correct multi-cloister typing by multicausal.

Line 204: Insert study on the impact of anemia in this age group and connect with your findings

Line 250: your date do not allow this conclusion.

General perception: discuss more about inflammation as it was associated.

5th Conclusions

Start the conclusions with the prevalence of anemia identified in the study, as this is in its title.

Review if after the new analyses if the conclusions will change.

References 23 and 27 – journal titles are not abbreviated.

Author Response

Many thanks for your valuable comments. All were attended and included in the final version. 

Reviewer 2 Report

The authors have performed a study of over 600 women of reproductive age in  a section of Cuba to determine the prevalence of iron deficiency and obesity/overweight/inflammation. They found a fair amount of anemia and iron deficiency and even more overweight. The results have public health implications for Cuba. Probably the results do not have widespread public health implications.

Comments:

The authors state that “The prevalence of anemia was 21.4%, iron storage deficiency at 16.0%, Anemia is associated with iron deposition deficiency OR=3.023 which was higher than the association of anemia with heavy menstrual bleeding (Anemia was associated with heavy menstrual bleeding OR=1.92). “ That is because even normal menstrual bleeding can lead to iron deficiency in the absence of sufficient iron in the diet. Therefore it is not logical to say . In conclusion, anemia is classified as a moderate public health problem, but not iron deficiency” because iron deficiency is only 16% and not 20%!

The authors state that “Iron storage deficiency was found in less than one-third of the cases.” I think that the problem is the fact that the authors used soluble transferrin receptor and not ferritin under 15 to declare the women “iron deficient” or had “absent iron stores”. If the  woman had a ferritin under 15 they classified her as “absent iron stores” and only if the soluble transferrin receptor was over 8.3 was the woman considered iron deficient. I appreciate the authors’ use of soluble transferrin receptor but really they could have used the ferritin to declare iron deficiency or not. There is now a growing body of literature that suggests that a ferritin of 15 is possibly too low a cutoff for iron deficiency but we can leave that aside for now, maybe women are iron deficient if their ferritin is 18 or 20 (or more).

In any case I am a practicing hematologist and not an epidemiologist and I do not agree with the way the authors interpreted the data. There are a lot of ways to try to diagnose iron deficiency with laboratory data and I don’t understand why they used soluble transferrin receptor and not transferrin saturation which is much more widely used. The literature suggests that the utility of soluble transferrin receptor is in situations where it is known that the individuals tested are sick such as in cystic fibrosis, HIV, active tuberculosis, rheumatoid arthrisis, hemodialysis. This is not the case here. I see it is widely used in epidemiological studies but it seems to me here that the results gave an incorrect estimation of the amount of true iron deficiency there is.

I also do not understand the cutoffs the authors cite in Table 1 which apparently were set for Cuba specifically. Why is the cutoff for what is considered a moderate public health problem so high? Personally I think that a prevalence of 5-19.9% of anemia is a HIGH prevalence, not a LOW prevalence. And I would not call anemia prevalence up to 39.9% is a MODERATE prevalence? That is quite forgiving of the authors, perhaps they did not wish to have it go on record that Cuba has a HIGH prevalence of anemia and iron deficiency.

It is possible that the authors have some competing interests here, maybe they are more concerned regarding overweight than iron deficiency? They are possibly looking to prove that inflammation which is a result of adiposity, causes anemia HOWEVER this is not an easy thing to establish. In particular not in women around 40 years old who are otherwise healthy. That is not actually a good population to study to prove that depression of erythropoiesis is due to inflammation.

The Discussion is not particularly well written. I don’t understand why they discuss zinc deficiency since that was not part of the study here. I also do not understand why they seem surprised that in Bangladesh, underweight status was associated with anemia and overweight was protective against anemia. That is similar to what was found here (lines 246-7). People without access to proper nutrition will be more likely to have iron deficiency. That is completely logical.

Lines 273-4 of Discussion are not correct since homocysteine is elevated in folic acid deficiency and they did study homocysteine levels.

Author Response

Many thanks for your valuable comments and contributions. All your comments were considered and included in the final version. 

Round 2

Reviewer 1 Report

The authors corrected practically all requests made. The article has greatly improved the quality and can be published. Thank you for the opportunity to review the article.